# Characterization of DNA Viruses in Hindgut Contents of *Protaetia brevitarsis* Larvae

**DOI:** 10.3390/insects16080800

**Published:** 2025-08-01

**Authors:** Jean Geung Min, Namkyong Min, Binh T. Nguyen, Rochelle A. Flores, Dongjean Yim

**Affiliations:** 1Department of Applied Biology, Kyungpook National University, Daegu 41566, Republic of Korea; jeanmin1@knu.ac.kr; 2Crop Protection Research Team, Department of Agricultural, Food and Environment Research, Gyeongsangbuk-do Agricultural Research & Extension Services, Daegu 41404, Republic of Korea; minnamkyong1@gmail.com; 3Hoxbio, Business Center, Gyeongsang National University, Jinju 52828, Republic of Korea; thanhbinhcnty@gmail.com; 4Faculty of Animal Science and Veterinary Medicine, Thai Nguyen University of Agriculture and Forestry, Thai Nguyen 24119, Vietnam; 5College of Veterinary Medicine & Institute of Animal Medicine, Gyeongsang National University, Jinju 52828, Republic of Korea; floresrochellea@gmail.com

**Keywords:** microbiota, phage, third-instar larvae, *Siphoviridae*, metagenome

## Abstract

Despite the known interaction between bacteria and viruses in the hindgut, little is known about the intestinal viral composition in larvae of the scarab species *Protaetia brevitarsis* (PBL). The viral profiles of PBL hindgut contents obtained from five farms located in different regions in Korea were investigated using metagenomic sequencing. This study confirmed that more than 98% of the DNA viruses in the hindgut were bacteriophages, mainly belonging to the *Siphoviridae* family. The remaining identified eukaryotic DNA viruses were unrelated or had little relation to insect viruses and likely originated from contaminated feed or soil. These results provide a comprehensive understanding of the gut viral composition of *P. brevitarsis* and will be helpful for studying the insect virome.

## 1. Introduction

Edible insects are generally considered an important source of high-quality proteins with a balanced amino acid profile, essential fatty acids, minerals, and other bioactive molecules, although the nutritional composition of insects varies considerably depending on the species, growth stage, rearing environmental conditions, and diet [1,2]. One of the main advantages of producing edible insects is that they help in producing livestock as feed and reduce overall livestock production [3]. The white-spotted flower chafer (*Protaetia brevitarsis*) is one of the edible insect species currently reared for food and feed due to its ease of production and high feed conversion efficiency [4,5]. *P. brevitarsis* is a species of beetle belonging to the Cetoniidae subfamily of the Coleoptera order. Although *P. brevitarsis* is considered an agricultural pest, its larvae have been long utilized in traditional medicine for centuries [6,7,8].

*P. brevitarsis*, similar to some other insects, can be reared using biological waste or organic matter as feed sources, and these feed sources can also be converted into high-value-added resources [9]. Potential agricultural biomass resources such as cotton stalks, maize straw, and sawdust can be converted into organic fertilizers with the larvae’s gut microbiota when used as feed for *Protaetia brevitarsis* larvae (PBL) [10,11,12]. The frass of the larvae, which has low heavy metal content, beneficial microorganisms, and varied nutrient elements, was tested as a fertilizer for the growth of ginseng sprouts [13]. In addition to its use in traditional medicine, *P. brevitarsis* has been used in animal feed, food, and biological waste recycling, and its frass can also be used as fertilizer. The microbial communities in the gut or frass of PBL have been analyzed, which could provide a foundation for in-depth research and applications of PBL resources [12,14,15].

Insects have been shown to commonly harbor varied microorganisms, including viruses, and they may serve as major reservoirs and vectors of infectious viruses, raising concerns about food and feed safety risks to human and animal health when insects are used as food and feed [16,17]. The gut viruses might also have positive or negative effects on animal health and bacterial community composition. There is a large amount of literature on the presence and roles of viruses, mainly in insects of economic or public health importance, such as silkworms and mosquitoes [16]. With advances in high-throughput sequencing technologies, diverse viruses in various insects have been discovered [18,19,20]. Liu et al. [19] indicated that environmental factors that shape and construct the viromes not only affect mosquito species. Bacteriophages also play a critical role in influencing the structure and function of the health gut microbiome and are primary drivers of bacterial fitness and diversity [21,22,23]. The virome composition can be affected by various factors, such as food/feed sources, geographical differences, microbial composition, and diseases [19,21,23]. The alteration of gut mucosal bacteriophages was demonstrated in ulcerative colitis in human subjects. An inverse correlation between the phage and its host genus of bacteria in ulcerative colitis was observed by virome analysis [21]. Our previous study indicated that the bacterial composition of the PBL hindgut had dynamics dependent on interactions between PBL and its environment [15]. Therefore, information on bacteriophages that affect the bacterial composition has become an important element of microbiota studies [21,23]. However, few studies have been performed on the virome in edible insects [16], including *P. brevitarsis*. Therefore, to obtain more information about the compositions of bacteriophages and eukaryotic viruses in the PBL hindgut, PBL collected from five farms in different regions in Korea were subjected to metagenomic shotgun sequencing in this study. Our results revealed that the abundance of viruses is likely more significantly shaped by environmental than genetic factors in composing the gut viral community in PBL.

## 2. Materials and Methods

### 2.1. Insects and Sampling

The hindgut samples were collected from the third-instar larvae of *P. brevitarsis* purchased from five specialized insect farms (KB, TO, BR, IS, JH) in three regions in Korea. KB and TO farms are located in Gyeongsangbuk-do province, Korea; BR and IS farms are located in Incheon province, Korea; and JH farm is located in Chungcheongnam-do province, Korea (Appendix A). Two farms (TO, JH) used their own feeds, and three farms (KB, BR, IS) used commercial feeds. The BR farm mainly used waste mushroom substrate, and the feeds of other four farms used oak sawdust as the main ingredient (Appendix A). Larvae were soaked for 1 min in 75% alcohol and washed three times with phosphate-buffered saline (PBS) [14,15]. The hindgut of the larvae was dissected, and approximately 500 mg of pooled hindgut contents from 10 larvae was collected in 2 mL sterile plastic tubes. The collected samples were stored at −80 °C until metagenomic shotgun sequencing was performed.

### 2.2. DNA Preparation and Metagenomic Shotgun Sequencing

Total metagenomic DNA was first extracted from 10 fecal samples using a fecal DNA isolation kit (MO BIO, Carlsbad, CA, USA) according to the manufacturer’s instructions. A whole genomic library was prepared using the TruSeq Nano DNA kit (Illumina, San Diego, CA, USA) by randomly fragmenting the DNA sample and then ligating it with specific adapters. The adapters used were AATGATACGGCGACCACCGAGATCTACAC[+library specific 8 sequences]ACACTCTTTCCCTACACGACGCTCTTCCGATCT and GATCGGAAGAGCACACGTCTGAACTCCAGTCAC[+library specific 8 sequences]ATCTCGTATGCCGTCTTCTGCTTG. Adapter-ligated fragments were amplified via PCR and then sequenced using the Illumina NovaSeq 6000 platform (Illumina, San Diego, CA, USA) and the 151 bp paired-end format (Macrogen, Seoul, Republic of Korea).

### 2.3. Bioinformatics and Sequencing Data Processing

Adapter sequences and data with an average Phred quality score of less than 20 were removed using Trimmomatic (v0.39) via the Kneaddata (v0.10.0) pipeline. To remove the *P. brevitarsis* genome sequence, reads mapped to the *P. brevitarsis* (GCA_004143645.1) reference genome were removed using Bowtie2 (v2.4.5). The compositional profiling of microbial communities (bacteria, archaea, and eukaryotes) was analyzed using MetaPhlAn4 (v4.0.0) and virus search (option: --mpa3 --add viruses). The reads were mapped to the specific marker genes of the microbial species using Bowtie2, and the species abundance was calculated based on the average number of reads mapped to the marker genes. At this point, marker genes mapped to less than 33% were removed because the species were considered to be absent. The β-diversity among the samples was estimated using the Bray–Curtis distance and visualized using principal coordinates analysis. Statistical significance between groups was assessed using the permutational analysis of variance (PERMANOVA) test via the QIIME 2 bioinformatics platform (v2020.8.0). Data were expressed as mean ± standard error values.

## 3. Results

### 3.1. General Information and Sample Sequencing

The hindgut contents of PBL were subjected to metagenomic shotgun sequencing, and the detailed sequencing data for each sample are presented (Appendix A). The sequencing generated an average of 82.6 M total reads (range: 78 to 86 M) and 37.6 M clean reads (range: 35.3 to 39.1 M). The average GC content and Q20 of these clean reads were 58.8 ± 1.1% and 97 ± 0.1%, respectively (Appendix A). Principal coordinate analysis based on taxonomy classification of metagenomic data was performed using the Bray–Curtis dissimilarity. Principal component 1 (PC1) and principal component 2 (PC2) explained 40.56% and 30.44% of the variance of the variables, respectively, explaining 71% of the total variance. The resulting plot showed that β-diversity was significantly different between the metagenomic data obtained from the five farms (PERMANOVA, pseudo-F = 46.95, *p* = 0.002) (Figure 1A). The taxonomic classification of metagenomic data revealed that the average bacteria and viruses were 86.4 ± 4.9% (range: 58.4 to 98.0) and 13.3 ± 4.9% (range: 1.3 to 41.5), respectively. Eukaryota was detected only in the BR farm and accounted for 0.29%, while Archaea was detected only in the KB farm and accounted for 1.11% (Figure 1B, Appendix A). These results indicated that viruses, along with bacteria, comprised a significant portion of the hindgut contents of PBLs, as well as that the abundance of viruses varied greatly between farms.

### 3.2. Analysis of Virus Composition

The results of the family-based taxonomic analysis showed that the relative abundance of viruses was approximately 41.2% of the metagenomic sequences identified at TO farm, followed by 15.0% at IS, 4.3% at BR, 4.0% at KB, and 1.6% at JH. The viruses identified were predominantly bacteriophages belonging to the *Siphoviridae*, *Podoviridae*, *Myoviridae*, *Tectiviridae*, and unclassified *Caudovirales* families, accounting for more than 98% of the hindgut DNA viruses. *Siphoviridae* was the most abundant viral family across all five farms, with relative abundances ranging from 1.66% to 40.54% (Figure 1C, Table 1). The other families had a prevalence of less than 1% of the gut microbiota. Viruses belonging to the *Podoviridae* family were detected at four farms (KB, TO, BR, and IS). Viruses belonging to the *Myoviridae* and *Tectiviridae* families were detected only at KB and IS farms and TO farm, respectively. Besides bacteriophages, a very small proportion of eukaryotic DNA viruses, accounting for 0.01–0.06% of the intestinal microbiota, was detected depending on the farms. Viruses belonging to the *Herpesviridae* family were detected on three farms (TO, BR, and IS), and viruses belonging to the *Retroviridae* and *Alloherpesviridae* families were detected on BR and TO farms, respectively (Table 1).

At the genus and species levels, taxonomic analysis indicated that the host bacteria of most phages were *Microbacterium*, *Mycobacterium*, and *Bacillus*. Phage Min1, which infected the genus *Microbacterium*, was detected at all farms, and it was found to be the most abundant bacteriophage, with a prevalence ranging from 0.9% to 29.09%. Phage viruses infecting *Streptomyces*, considered agriculturally beneficial bacteria [14], were detected at three farms (KB, BR, IS). Phage viruses infecting *Escherichia coli* were not abundant, having prevalences of 0.1% at the KB farm and 0.03% at the IS farm. Only five eukaryotic DNA viruses were detected: human endogenous retrovirus K (HERV-K), equid alphaherpesvirus 4 (EHV-4), bovine alphaherpesvirus 5 (BoHV-5), ateline gammaherpesvirus 3 (AtHV-3), and anguillid herpesvirus 1. The prevalence of these eukaryotic DNA viruses was very low, ranging from 0.01% to 0.06%, and they were found at only three farms (TO, BR, IS) (Table 1 and Appendix A).

## 4. Discussion

Enteric viruses play varied roles in insect health and homeostasis, including immunity, digestion, nutrient absorption, and interactions with other gut microbes [24,25]. The intestinal viral composition is a dynamic and evolving community influenced by factors such as diet, environmental conditions, and the immune status of the host [24,25]. Advances in high-throughput sequencing technologies have greatly improved our ability to understand the complexity of the gut microbiome in various species over the past decade. The diversity and abundance of either DNA or RNA viruses in insects and other animals have been monitored via metagenomics approaches [19,20]. Phages account for more than 90% of the human gut virome, of which DNA phages make up the majority, while RNA phages are much less abundant. In eukaryotic viruses, most eukaryotic DNA viruses are latent or dormant under normal conditions. Eukaryotic RNA viruses are rare under healthy conditions, and most insects, eukaryotic RNA viruses are plant viruses [24,26]. Therefore, first of all, this study demonstrated the diversity and abundance of DNA viruses present in the hindgut contents of PBL obtained from five farms through metagenomic shotgun analysis.

In this study, the viral abundance in the PBL hindgut microbiota varied greatly between farms, ranging from 6% to 41.2%, and, thus, β-diversity using metagenomic data also showed significant differences. Similarly, our previous study analyzed the hindgut microbiota of PBL third-instar larvae collected from five farms and found that there were significant differences in bacterial microbiota between farms. Min et al. [15] found environmental conditions such as temperature, humidity, and ventilation to be more important in shaping the gut microbiota than feed ingredients or genetic predisposition. Most of the viruses identified in the PBL hindgut in this study were bacteriophages, accounting for more than 98% of hindgut DNA viruses, mainly belonging to the *Siphoviridae* family. On the other hand, eukaryotic DNA viruses, accounting for 0.01–0.06% of the gut microbiota, were likely to have originated from non-insect-related animals such as humans, horses, cows, and eels. Eukaryotic DNA viruses that infect insects such as coleopteran species, orthopteran species, lepidopteran species, and dipteran species mainly belong to the families *Baculoviridae*, *Herpesviridae*, *Nudiviridae*, *Entomopoxviridae*, *Iridoviridae*, *Parvoviridae*, and *Asfarviridae*. These viruses can have a serious impact on insect populations, causing disease and mortality [16,17]. Interestingly, no virus has been detected in the wax moth, *Achroia grisella*, so far [16]. Collectively, these results suggest that eukaryotic DNA viruses detected via metagenomic shotgun analysis of hindgut contents of PBL may have originated from contaminated feed or soils. Therefore, further studies are needed to determine whether *P. brevitarsis* adults can be infected with eukaryotic DNA viruses or form a symbiotic relationship with them.

The most abundant phage family in public databases is *Siphoviridae* (66%), followed by *Myoviridae* (20%) and *Podoviridae* (14%) [26]. Similarly, in our study, the most prevalent viruses in the gut microbiota belonged to the *Siphoviridae*, followed by the *Podoviridae*. At the species level, Min1 phages infecting the *Microbacterium* genus, making up Gram-positive, nonspore-forming, rod-shaped bacteria, were found to be the most abundant bacteriophage in intestinal DNA viruses. The Min1 phage was detected at all farms and showed a prevalence ranging from 0.9% to 29.09% in gut microbiota. However, in our study and previous studies, the genus *Microbacterium* was not included among the top 27 most abundant genera in the PBL hindgut microbiota analysis [15] or among the top 20 most abundant genera in the PBL frass microbiota analysis [14]. Most *Microbacterium* species appear to be soil organisms specializing in decomposing complex organic substrates [27]. Generally, organic substrates such as oak sawdust and spent mushroom substrate were used as PBL feed after processing [28]. Taken together, these results suggest that PBL may have ingested Min1 phages present in the soil, although information on the given metagenomic-derived sequences is not sufficient to confirm the presence of replicating DNA viruses.

For studying the widespread use of *P. brevitarsis*, studies on *P. brevitarsis* itself are required. Therefore, transcriptome and genome analyses, as well as intestinal and frass bacterial microbiota analyses, of *P. brevitarsis* were previously conducted [14,15,29,30]. The intestinal tract contains various microbiota, including viruses, and plays a critical role in animal health and homeostasis. Changes in the composition of the gut microbiota of housefly larvae via phage treatment were shown to negatively affect the growth and development of housefly larvae [25], indicating that the composition of viruses is important for the insect industry. This study focused on profiling DNA viruses in the hindgut contents of PBL using metagenomic shotgun sequences. It was confirmed that more than 98% of DNA viruses in the hindgut were bacteriophages, mainly belonging to the *Siphoviridae* family. The detected eukaryotic DNA viruses were thought to have likely originated from contaminated feed or soil. These results provide a comprehensive understanding of the gut DNA viral composition of PBL and will aid in future studies of insect viromes.

## Figures and Tables

**Figure 1 insects-16-00800-f001:**
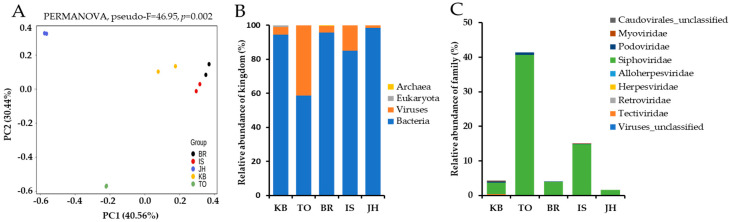
The diversity and composition of the hindgut microbiota of *P. brevitarsis* larvae. (**A**) The β-diversity based on metagenomics data, visualized using multidimensional scaling on the Bray–Curtis distance. *p*-value derived from the PERMANOVA test with 999 permutations. (**B**) Each column represents the average relative abundance of each kingdom of the microbial community. (**C**) The relative abundance of the viral family. The community plots indicate percentage abundance at different taxonomic levels. KB, Kyungpook farm; TO, Tohamsan Gumbengi farm; BR, Gumbengi brothers; IS, Secomnalagum Gumbengi; JH, Jhbio; PC, principal component.

**Table 1 insects-16-00800-t001:** A taxonomy analysis of hindgut viruses of *P. brevitarsis* larvae at the family and genus levels.

Farm Name			Relative Prevalence (%)			
		KB	TO	BR	IS	JH
Family	Viruses_unclassified	0.14	-	-	0.05	-
	*Tectiviridae*	0.22	-	-	-	-
	*Retroviridae*	-	-	0.06	-	-
	*Herpesviridae*	-	0.05	0.01	0.02	-
	*Alloherpesviridae*	-	0.05	-	-	-
	*Siphoviridae*	3.41	40.54	3.89	14.72	1.66
	*Podoviridae*	0.30	0.74	0.13	0.20	-
	*Myoviridae*	0.19	-	-	0.09	-
	*Caudovirales*_unclassified	0.10	-	-	-	-
	Other (Bacteria, Archaea, Eukaryota)	95.64	58.61	95.92	84.91	98.34
Genus	Viruses_unclassified	0.14	-	-	0.05	-
	*Betatectivirus*	0.22	-	-	-	-
	*Retroviridae*_unclassified	-	-	0.06	-	-
	*Varicellovirus*	-	0.05	0.01	-	-
	*Rhadinovirus*	-	-	-	0.02	-
	*Cyprinivirus*	-	0.05	-	-	-
	*Yuavirus*	0.03	-	-	-	-
	*Siphoviridae*_unclassified	2.64	33.82	2.53	6.55	0.90
	*Send513virus*	-	-	-	0.03	0.01
	*Phic31virus*	0.32	-	0.05	0.14	-
	*Pbi1virus*	0.22	1.04	0.43	0.21	0.15
	*Omegavirus*	-	-	-	0.01	-
	*L5virus*	-	-	0.61	6.79	0.15
	*Fishburnevirus*	-	1.68	-	0.05	0.02
	*Che9cvirus*	0.12	0.70	0.10	0.63	0.14
	*Che8virus*	0.08	2.38	0.16	0.24	0.21
	*Brujitavirus*	-	0.93	-	0.07	0.08
	*Podoviridae*_unclassified	-	-	0.10	-	-
	*Bpp1virus*	0.30	0.74	0.02	0.20	-
	*Svunavirus*	0.19	-	-	-	-
	*P1virus*	-	-	-	0.03	-
	*Bxz1virus*	-	-	-	0.06	-
	*Caudovirales*_unclassified	0.10	-	-	-	-
	Other (Bacteria, Archaea, Eukaryota)	95.64	58.61	95.92	84.92	98.33

## Data Availability

The data presented in this study are available in the NCBI Sequence Read Archive, PRJNA1264687.

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
