# Peer review of "Characterization of DNA Viruses in Hindgut Contents of Protaetia brevitarsis Larvae"

_insects, 2025, doi:10.3390/insects16080800_

Round 1
Reviewer 1 Report
Comments and Suggestions for Authors
This manuscript addresses a relatively underexplored topic—characterization of the gut DNA virome in Protaetia brevitarsis larvae (PBL)—using a metagenomic approach.
Lines 12–13 Consider clarifying the "interdependence" early. What kind of interactions—nutritional, competitive, or host-mediated—are implied? Or just simply say "Despite the known interactions between bacteria and viruses in insect hindguts..."
Line 21–23 The phrase "traditional animal medicines" is confusing. Do you mean “used in traditional medicine and as animal feed”?
Line 31–33 "with approximately 41.2% (TO), 15% (IS)..." It's unclear what the percentages refer to—total viral reads or overall metagenome? Also, please define the abbreviations for farms (TO, IS, BR, etc.) at first mention.
Suggestion: "...with viral reads comprising approximately 41.2%, 15.0%, 4.3%, 4.0%, and 1.6% of the metagenomic sequences from Farms TO, IS, BR, KB, and JH, respectively. (Farm codes are abbreviated; detailed locations are provided in the Methods section.)"
Lines 56–58 better refine the sentence e.g. “Although P. brevitarsis is considered an agricultural pest, its larvae have long been utilized in traditional medicine...”
Line 76–78 “bacteria composition” should be “ bacterial composition”
Line 83 "may be mainly influenced by environmental factors..." change to “are likely shaped more by environmental than genetic factors.” But give some references to this point.
Line 89 Please clarify whether this weight was from each pooled sample (i.e., 10 larvae combined) or from each larva individually. Consider rephrasing to: “Approximately 500 mg of pooled hindgut contents from 10 larvae were collected…”
Lines 110–113: While the use of Bray–Curtis distance and PERMANOVA is appropriate for assessing β-diversity differences between samples, I recommend considering ANOSIM as a complementary test. ANOSIM is robust to differences in dispersion between groups and could help confirm the observed group-wise differences in viral community composition. Also, please clarify whether UniFrac was indeed used and how phylogenetic information was obtained for shotgun-derived viral data—otherwise, consider removing the UniFrac reference.
L127–132: Say “Bray–Curtis dissimilarity” instead of “distance statistic.” ; The explanation of PC1 and PC2 is good, but the phrase “cumulative contribution rate of up to 71%” could be simplified to “explaining 71% of total variance.”; Good use of PERMANOVA, but the pseudo-F value should include degrees of freedom if available (pseudo-Fâ‚„,â‚‚â‚€ or similar).
L149: “rate of viruses” → “relative abundance of viruses”
L150–151: Clarify: “Viruses made up approximately 41.2% of the microbiota in TO farm, followed by 15.0% in IS, 4.3% in BR, 4.0% in KB, and 1.6% in JH.”
L154–156: Instead of repeating “in the intestinal microbiota” twice, write:
“Siphoviridae was the most abundant viral family across all farms, with relative abundances ranging from 1.66% to 40.54% (Figure 1C, Table 1).”
In addition, The introduction in this manuscript provides a basic background, but it lacks depth and could be significantly improved to better justify the study.
- It does not adequately describe prior work on insect-associated viruses (either in beetles or other models like mosquitoes, termites, etc.). No references are provided to support statements about the roles of viruses in gut ecosystems (e.g., phage-mediated microbial modulation, horizontal gene transfer, or effects on host immunity).Suggestion: Add citations to key virome studies in insects or broader gut ecosystems (e.g., phage roles in shaping microbiota or ecological stability), such as:Zuo et al., 2019, Gut phageome and health; Manrique et al., 2016, Gut phageome and dysbiosis; Studies on viral metagenomics in insect guts (e.g., termites, Drosophila, mosquitoes).
- While the study is descriptive, even a short hypothesis (e.g., viral diversity may vary due to diet/feed source, regional differences, or microbial composition) would add value.
- The authors mention concepts like the roles of viruses in microbiomes and the importance of microbial diversity in insect guts—but don’t cite any background work to back these statements. Because the study is exploratory and descriptive, anchoring it with just a few key metagenomic or insect virome studies would strengthen credibility without adding length.
Author Response
We sincerely appreciate the editor and reviewers for their valuable time, and insightful recommendations for enhancing our manuscript. The authors have diligently considered the comments, and below are our point-for-point responses. We trust these will align with your expectations and contribute to the further improvement of our manuscript. Authors edited English also.
Reviewer’s comments 1:
This manuscript addresses a relatively underexplored topic—characterization of the gut DNA virome in Protaetia brevitarsis larvae (PBL)—using a metagenomic approach.
Lines 12–13 Consider clarifying the "interdependence" early. What kind of interactions—nutritional, competitive, or host-mediated—are implied? Or just simply say "Despite the known interactions between bacteria and viruses in insect hindguts..."
Response: Changed with the suggestion in line 14.
Line 21–23 The phrase "traditional animal medicines" is confusing. Do you mean “used in traditional medicine and as animal feed”?
Response: Changed with the suggestion in line 24.
Line 31–33 "with approximately 41.2% (TO), 15% (IS)..." It's unclear what the percentages refer to—total viral reads or overall metagenome? Also, please define the abbreviations for farms (TO, IS, BR, etc.) at first mention.
Suggestion: "...with viral reads comprising approximately 41.2%, 15.0%, 4.3%, 4.0%, and 1.6% of the metagenomic sequences from Farms TO, IS, BR, KB, and JH, respectively. (Farm codes are abbreviated; detailed locations are provided in the Methods section.)"
Response: Full names of abbreviations used for farms have been added. The indicated sentence was modified as “Family-based taxonomic analysis indicated that the relative abundance of viruses in the gut overall metagenome varied significantly between farms, with viral reads comprising approximately 41.2%, 15.0%, 4.3%, 4.0%, and 1.6% of metagenomic sequences from farms Tohamsan gumbengi farm (TO), Secomnalagum gumbengi (IS), Gumbengi brothers (BR), Kyungpook farm (KB), and Jhbio (JH), respectively” in lines 34-38.
Detailed locations of farms are provided as “KB and TO farms are located in Gyeongsangbuk-do province, BR and IS farms are located in Incheon province, and JH farm is located in Chungcheongnam-do province, korea” in the section of materials and methods in lines 104-105.
Lines 56–58 better refine the sentence e.g. “Although P. brevitarsis is considered an agricultural pest, its larvae have long been utilized in traditional medicine...”
Response: The indicated sentence was modified as “Although P. brevitarsis is considered an agricultural pest, its larvae have long been utilized in traditional medicine for centuries” in lines 60-62.
Line 76–78 “bacteria composition” should be “bacterial composition”
Response: Indicated error was corrected as “bacterial community composition” in line 78.
Line 83 "may be mainly influenced by environmental factors..." change to “are likely shaped more by environmental than genetic factors.” But give some references to this point.
Response: The indicated sentence was modified as “Our results revealed that the abundance of viruses is likely more significantly shaped by environmental than genetic factors in composing the gut viral community in PBL.” in line 97. Because this sentence describes our research results, it is difficult to add references.
Line 89 Please clarify whether this weight was from each pooled sample (i.e., 10 larvae combined) or from each larva individually. Consider rephrasing to: “Approximately 500 mg of pooled hindgut contents from 10 larvae were collected…”
Response: The indicated sentence was modified as “approximately 500 mg of pooled hindgut contents from 10 larvae was collected in 2 ml sterile plastic tubes” in line 110.
Lines 110–113: While the use of Bray–Curtis distance and PERMANOVA is appropriate for assessing β-diversity differences between samples, I recommend considering ANOSIM as a complementary test. ANOSIM is robust to differences in dispersion between groups and could help confirm the observed group-wise differences in viral community composition. Also, please clarify whether UniFrac was indeed used and how phylogenetic information was obtained for shotgun-derived viral data—otherwise, consider removing the UniFrac reference.
Response: The indicated sentence “Unweighted UniFrac distances (b-diversity values) were calculated based on pre-calculated phylogenetic trees in MetaPhlAn4 (v4.0.0)” was deleted.
Consistent with the results of PERMANOVA, ANOSIM (Analysis of similarities) using the Bray-Curtis distance indicated the p-value of 0.002 with 999 permutations.
L127–132: Say “Bray–Curtis dissimilarity” instead of “distance statistic.”; The explanation of PC1 and PC2 is good, but the phrase “cumulative contribution rate of up to 71%” could be simplified to “explaining 71% of total variance.”; Good use of PERMANOVA, but the pseudo-F value should include degrees of freedom if available (pseudo-Fâ‚„,â‚‚â‚€ or similar).
Response: The indicated sentences were modified according to the suggestion in lines 147 and 149, respectively. The sentence “β-diversity was significantly different between metagenomic data obtained from the five farms (PERMANOVA, pseudo-F = 46.95, p = 0.002)” was added in lines 150-151. In addition, the information on “pseudo-F = 46.95” was added in Figure 1A.
L149: “rate of viruses” → “relative abundance of viruses”
Response: “rate of viruses” was changed to “the relative abundance of viruses” in line 168.
L150–151: Clarify: “Viruses made up approximately 41.2% of the microbiota in TO farm, followed by 15.0% in IS, 4.3% in BR, 4.0% in KB, and 1.6% in JH.”
Response: The related sentence was changed as “The results of the family-based taxonomic analysis showed that the relative abundance of viruses was approximately 41.2% of metagenomic sequences in TO farm, followed by 15.0% in IS, 4.3% in BR, 4.0% in KB, and 1.6% in JH” in lines 168-170.
L154–156: Instead of repeating “in the intestinal microbiota” twice, write:
“Siphoviridae was the most abundant viral family across all farms, with relative abundances ranging from 1.66% to 40.54% (Figure 1C, Table 1).”
Response: The indicated sentence was changed with the suggested sentence in lines 173-174.
In addition, the introduction in this manuscript provides a basic background, but it lacks depth and could be significantly improved to better justify the study.
Q1. It does not adequately describe prior work on insect-associated viruses (either in beetles or other models like mosquitoes, termites, etc.). No references are provided to support statements about the roles of viruses in gut ecosystems (e.g., phage-mediated microbial modulation, horizontal gene transfer, or effects on host immunity).Suggestion: Add citations to key virome studies in insects or broader gut ecosystems (e.g., phage roles in shaping microbiota or ecological stability), such as:Zuo et al., 2019, Gut phageome and health; Manrique et al., 2016, Gut phageome and dysbiosis; Studies on viral metagenomics in insect guts (e.g., termites, Drosophila, mosquitoes).
Response: The added sentences are “With advances in high-throughput sequencing technologies, diverse viruses in various insects have been discovered (18-20). Liu et al. (19) indicated that environmental factors that shape and construct the viromes do not only affect mosquito species. Bacteriophages also play a critical role in influencing the structure and function of health gut microbiome and are primary drivers of bacterial fitness and diversity (21-23). The virome composition can be affected by various factors, such as food/feed sources, geographical differences, microbial composition, and diseases (19, 21, 23). The alteration of gut mucosal bacteriophages was demonstrated in ulcerative colitis in human subjects. An inverse correlation between the phage and its host genus bacteria in ulcerative colitis was observed by virome analysis (21)” in lines 80-89
Q2. While the study is descriptive, even a short hypothesis (e.g., viral diversity may vary due to diet/feed source, regional differences, or microbial composition) would add value.
Response: The related information was added as “The virome composition can be affected by various factors, such as food/feed sources, geographical differences, microbial composition, and diseases (19, 21, 23)” in lines 85-86.
Q3. The authors mention concepts like the roles of viruses in microbiomes and the importance of microbial diversity in insect guts—but don’t cite any background work to back these statements. Because the study is exploratory and descriptive, anchoring it with just a few key metagenomic or insect virome studies would strengthen credibility without adding length.
Response: The related information was added as “Therefore, information on bacteriophages that affect the bacterial composition has become an important element of microbiota studies (21, 23)” in lines 91-93.
Reviewer 2 Report
Comments and Suggestions for Authors
This is an interesting article which investigates the presence of DNA viruses in Protaetia brevitarsis larvae, an insect of interest due to its use in food more recently. The study uses a small number of different farms (5) taking ten insects from each site to assess DNA viral carriage within the gut of the insects.
For me, the number of insects and farms is too small, and there is no evidence of any controls included within the study to assess for contamination during DNA isolation.
There is also a lot of discussion around bacteria within the samples, but this is not presented. Indeed it is very similar to a previous study by the same authors on the bacteria within the gut of the same insect species.
Line 63- please define PBL in the main text, although I appreciate that it is defined in the abstract
Line 73-74- not sure that I follow this bit, please consider rewording
Similar with 77-79- please reword as this doesn’t make sense? Maybe was shown to be highly dynamic is what you are trying to say?
Line 86- is it possible to provide more information on the farms please? Like what they were fed, numbers of insects etc? any biosecurity practises? This is line 119-122- and maybe worth moving to the methods, but I will leave this to the discretion of the authors
Were any controls included during DNA extractions? And sequencing?
Section 3.1. You talk about the bacteria in here but don’t present it. Is there any benefit in presenting the bacterial data alongside the viral data? Again not a requirement, more a thought question
Line 171- this should be bacteria rather than bacterium, or an agriculturally beneficial bacterium
Line 230-243- this doesn’t seem to be a conclusion and most of this may benefit from being included in the discussion
The study shows stark similarities to the previous studies from the group ‘Min, N.; Min, J.G.; Cammayo-Fletcher, P.L.T.; Nguyen, B.T.; Yim, D. Comparative Analysis of Hindgut Microbiota Variation in Protaetia brevitarsis Larvae across Diverse Farms. Microorganisms 2024, 12, 496. https://doi.org/10.3390/microorganisms12030496’ Indeed it uses the same farms. My biggest question is thus why was the DNA virus stuff not included with the bacterial stuff in this paper? And are we then likely to see another paper entitled RNA viruses from Protaetia brevitarsis larvae?
Author Response
We sincerely appreciate the editor and reviewers for their valuable time, and insightful recommendations for enhancing our manuscript. The authors have diligently considered the comments, and below are our point-for-point responses. We trust these will align with your expectations and contribute to the further improvement of our manuscript.
This is an interesting article which investigates the presence of DNA viruses in Protaetia brevitarsis larvae, an insect of interest due to its use in food more recently. The study uses a small number of different farms (5) taking ten insects from each site to assess DNA viral carriage within the gut of the insects.
For me, the number of insects and farms is too small, and there is no evidence of any controls included within the study to assess for contamination during DNA isolation.
Response: Two samples were collected from each farm. One sample was the hindgut contents of 10 larvae, which were thoroughly mixed before DNA extraction. Metagenome sequencing requires very small sample volumes. Therefore, the authors believed that sampling 20 larvae would provide sufficient information about the metagenome of each farm. Future studies will increase the number of insects to minimize variations between samples. The number of farms used in the experiment is estimated to be approximately 10% of the total farms in Korea, and the number of farms will be increased in future experiments as the suggestion. Although the control DNA was not included in the samples, the larvae to remove any possible contamination were soaked in 75% alcohol during 1 minute and washed three times with phosphate-buffered saline (PBS). In addition, hindgut contents were collected on a clean bench using a disposable plastic container and an autoclaved scalpel for each larva. DNA extraction was performed using an automated machine.
There is also a lot of discussion around bacteria within the samples, but this is not presented. Indeed, it is very similar to a previous study by the same authors on the bacteria within the gut of the same insect species.
Response: The results obtained from the metagenomic sequence included a variety of bacteriophages that were closely related to the abundance and composition of the bacteria. To date, only two or three papers have been published on the microbiome analysis of Protaetia brevitarsis, and no papers have been published on the virus analysis based on our knowledge. Additionally, the authors used Protaetia brevitarsis larvae collected from the same farm for metagenomic analyses. Figure 1B contains information on the relative abundance of bacteria. Therefore, some of the discussion may be similar to our previous paper, and parts of our previous paper have been referenced. However, in future studies, we will analyze more extensive data beyond the limited information and conduct a detailed analysis.
Line 63- please define PBL in the main text, although I appreciate that it is defined in the abstract
Response: The sentence was modified as “Protaetia brevitarsis larvae (PBL)” in line 67.
Line 73-74- not sure that I follow this bit, please consider rewording
Similar with 77-79- please reword as this doesn’t make sense? Maybe was shown to be highly dynamic is what you are trying to say?
Response: The sentence was modified as “The gut viruses might also have positive or negative effects on animal health and bacterial community composition” in lines 77-78. Also, the indicated sentence was modified as “Our previous study indicated that the bacterial composition of PBL hindgut had dynamics dependent on interactions between PBL and its environment (15)” in lines 89-91.
Line 86- is it possible to provide more information on the farms please? Like what they were fed, numbers of insects etc? any biosecurity practises? This is line 119-122- and maybe worth moving to the methods, but I will leave this to the discretion of the authors
Response: The additional sentence was added with “KB and TO farms are located in Gyeongsangbuk-do province, Korea; BR and IS farms are located in Incheon province, Korea; and JH farm is located in Chungcheongnam-do province, Korea” in lines 104-105. Also, the indicated sentence was moved to the section 2.1. of Materials and Methods.
Were any controls included during DNA extractions? And sequencing?
Response: Although the control DNA was not included in the samples, the larvae to remove any possible contamination were soaked in 75% alcohol during 1 minute and washed three times with phosphate-buffered saline (PBS). In addition, hindgut contents were collected on a clean bench using a disposable plastic container and an autoclaved scalpel for each larva. DNA extraction was performed using an automated machine and extracted metagenome was sequenced by an Illumina NovaSeq 6000 platform.
The additional sentence was added as “The adapters used were AATGATACGGCGACCACCGAGATCTACAC[+library specific 8 sequences] ACACTCTTTCCCTACACGACGCTCTTCCGATCT and GATCGGAAGAGCACACGT-CTGAACTCCAGTCAC[+library specific 8 sequences]ATCTCGTATGCCGTCTTCTGCTTG. Adapter-ligated fragments were amplified via PCR and then sequenced using the Illumina NovaSeq 6000 platform and the 151 bp paired-end format (Macrogen, Seoul, Korea)” in lines 118-124.
Section 3.1. You talk about the bacteria in here but don’t present it. Is there any benefit in presenting the bacterial data alongside the viral data? Again not a requirement, more a thought question
Response: Figure 1B contains information on the relative abundance of bacteria obtained during metagenomic seqeucing. Therefore, the authors simply described the proportion of bacteria in lines 152-154 as “The taxonomic classification of metagenomic data revealed that the average bacteria and viruses were 86.4 ± 4.9% (range: 58.4 to 98.0) and 13.3 ± 4.9% (range: 1.3 to 41.5), respectively” Because the composition of intestinal bacteria and the composition of bacteriophages are closely related, it would be better to provide information on them together.
Line 171- this should be bacteria rather than bacterium, or an agriculturally beneficial bacterium
Response: The indicated sentence was modified with “agriculturally beneficial bacteria” in line 189.
Line 230-243- this doesn’t seem to be a conclusion and most of this may benefit from being included in the discussion
Response: The conclusion section was deleted. The conclusion section has been moved to the end of the discussion in the revised version.
The study shows stark similarities to the previous studies from the group ‘Min, N.; Min, J.G.; Cammayo-Fletcher, P.L.T.; Nguyen, B.T.; Yim, D. Comparative Analysis of Hindgut Microbiota Variation in Protaetia brevitarsis Larvae across Diverse Farms. Microorganisms 2024, 12, 496. https://doi.org/10.3390/microorganisms12030496’ Indeed it uses the same farms. My biggest question is thus why was the DNA virus stuff not included with the bacterial stuff in this paper? And are we then likely to see another paper entitled RNA viruses from Protaetia brevitarsis larvae?
Response: The authors used the same batch of samples stored in a deep freezer for metagenomic sequencing. For bacteria, the relative abundance of bacteria is represented by blue bars in Figure 1B. Unlike insects such as mosquitoes, the insect studied with Protaetia brevitarsis has no or very few identified euakaryotic DNA viruses, so future studies will analyze a wide range of RNA viruses. Thank you for your valuable suggestion.
Reviewer 3 Report
Comments and Suggestions for Authors
Min et al. report the results of a laboratory study on the presence of DNA viruses in the hindgut of scarab larvae.
The study requires major improvement.
MAJOR POINTS
(I) I was unable to determine whether this study was replicated or not.
(II) The clarity of the text requires attention at many points.
I have written numbered points on a scanned copy of the manuscript.
NUMBERED POINTS (see scanned file)
1. The authors may consider including the family and order of the insect in the title (just a suggestion).
2. Different regions? Of Korea? Or the world?
3. Unclear what the initials mean in the summary - better just to give the range of percentages.
4. Please report the findings in the past tense.
5. microflora? Better microbiota.
6. Surely the environment has a large contribution.
7. Do not use keywords that are already in the title.
8. Edible insects are not really "important" sources of proteins are they......they might be in the future. Soybean, beef or fish are important sources.
9. Edible insects... reduce overall livestock production? Please explain how.
10. Cetoniinae is a subfamily, isn't it? Most people know this family as Scarabaeidae.
11a. I think that it's important to mention that there is NO evidence of harmful human viruses infecting insects, to date, as far as I am aware. This text needs to be modified to make this clear.
11b. Regions? Of Korea?
12. Farms? Were these agricultural farms? Or specifically insect rearing facilities?
13. For how long were larvae soaked in ethanol?
14. What kit was used exactly?
15. Please provide amplification and primers details.
16. How many samples were analyzed? Each sample was taken from 10 insects. But how many samples did you sequence? Was this study replicated?
17. Do you mean "among" samples?
18. How many replicates were analyzed by t test?
19. This text should be moved to section 2.1 (methods)
20. Are the ± values SE? SD?
21. occupy? Do you mean represented? or comprised?
22. rate of viruses? Do you mean "prevalence based on the numbers of amplified reads"?
23. Virus families are written in italics.
24. Table 1. Are these values prevalences (%)? Not stated.
25. Genera are written in italics.
26. Provide a reference here.
27. Are you specifically talking about insect guts here?
28. There are published studies on DNA viruses in scarab larvae and in other coleopteran larvae. Mention them. Reference 16 is not a useful source here.
29. You performed this study and identified an abundance of phages, so what are your conclusions from the production standpoint? Are phages of benefit or detrimental to the production of scarab larvae?
30. Delete repetitive text. This is not a conclusion (its repeating preamble)
31. This text is Discussion, not a conclusion.
32. This text does look like a conclusion, but what are the implications for scarab production on a commercial scale?
33. Why is working for Hoxbio a potential conflict of interest? Does this company produce scarabs? Or what?
34. The references should be formatted following MDPI guidelines.
Other points
Supplemental Tables.
S1: are the farms in Korea? The BR diet is described as mushroom in the table, but is described as mushroom waste in the text. Which is it?
S3. Indicate what the values mean? Are these percentages? Of what? The term "Microbial community " is not informative.
S4. Again - what do the values indicate exactly? "Taxonomy analysis" is not informative.

Requires improvement (see my suggestions).
Author Response
We sincerely appreciate the editor and reviewers for their valuable time, and insightful recommendations for enhancing our manuscript. The authors have diligently considered the comments, and below are our point-for-point responses. We trust these will align with your expectations and contribute to the further improvement of our manuscript. Also, authors edited English.
Min et al. report the results of a laboratory study on the presence of DNA viruses in the hindgut of scarab larvae.
The study requires major improvement.
MAJOR POINTS
(I) I was unable to determine whether this study was replicated or not.
Response: Protaetia brevitarsis larvae were obtained from five different farms, and two samples were collected from each farm. One sample was the hindgut contents of 10 larvae, which were thoroughly mixed before DNA extraction. Metagenome sequencing requires very small sample volumes. Therefore, the authors believed that sampling 20 larvae would provide sufficient information about the metagenome of each farm.
(II) The clarity of the text requires attention at many points.
1. The authors may consider including the family and order of the insect in the title (just a suggestion).
Response: Authors would like to keep the title.
- Different regions? Of Korea? Or the world?
Response: “in Korea” was added in line 17. - Unclear what the initials mean in the summary - better just to give the range of percentages.
Response: The indicated sentence was changed as “Family-based taxonomic analysis indicated that the relative abundance of viruses in the gut overall metagenome varied significantly between farms, with viral reads comprising approximately 41.2%, 15.0%, 4.3%, 4.0%, and 1.6% of metagenomic sequences from the farms Tohamsan gumbengi farm (TO), Secomnalagum gumbengi (IS), Gumbengi brothers (BR), Kyungpook farm (KB), and Jhbio (JH), respectively” in lines 34-38. - Please report the findings in the past tense.
Response: English of the revised version was edited by MDPI. The sentences were changed as past tense. - microflora? Better microbiota.
Response: Changed with “microbiota” in line 43. - Surely the environment has a large contribution.
Response: Yes, environment factors are a critical factor for bacterial community and phage composition. - Do not use keywords that are already in the title.
Response: Keywords were changed as “microbiota; phage; third-instar larvae; Siphoviridae; metagenome” - Edible insects are not really "important" sources of proteins are they......they might be in the future. Soybean, beef or fish are important sources.
Response: In developing and underdeveloped countries, edible insects can be an important source of protein. Insects can also be used as a source of protein in livestock feed. - Edible insects... reduce overall livestock production? Please explain how.
Response: The sentence was annotated of published paper (3). It is difficult at present, but it is possible if processing capabilities for insects for flavor, etc. are developed. Temperatures are continuously rising, and protein production using insects is quite cost-effective, so it is also considered possible. - Cetoniinae is a subfamily, isn't it? Most people know this family as Scarabaeidae.
Response: The indication was modified as “Cetoniidae subfamily” in line 60.
11a. I think that it's important to mention that there is NO evidence of harmful human viruses infecting insects, to date, as far as I am aware. This text needs to be modified to make this clear.
Response: The sentence was modified as “Insects have been shown to commonly harbor varied microorganisms, including viruses, and they may serve as major reservoirs and vectors of infectious viruses” in line 75.
11b. Regions? Of Korea?
Response: “in Korea” was added in line 96.
- Farms? Were these agricultural farms? Or specifically insect rearing facilities?
Response: “five specialized insect farms” was added in line 103. - For how long were larvae soaked in ethanol?
Response: The indicated sentence was modified as “Larvae were soaked for 1 min in 75% alcohol” in line 109. - What kit was used exactly?
Response: The related information was added as “using a fecal DNA isolation kit (MO BIO, Carlsbad, CA, USA)” in line 116. - Please provide amplification and primers details.
Response: The information was added as “The adapters used were AATGATACGGCGACCACCGAGATCTACAC[+library specific 8 sequences]ACACTCTTTCCCTACACGACGCTCTTCCGATCT and GATCGGAAGAGCACACGTCTGAACTCCAGTCAC[+library specific 8 sequences ]ATCTCGTATGCCGTCTTCTGCTTG.” in lines 118-121. - How many samples were analyzed? Each sample was taken from 10 insects. But how many samples did you sequence? Was this study replicated?
Response: Protaetia brevitarsislarvae were obtained from five different farms, and two samples were collected from each farm. One sample was the hindgut contents of 10 larvae, which were thoroughly mixed before DNA extraction. Metagenome sequencing requires very small sample volumes. Therefore, the authors believed that sampling 20 larvae would provide sufficient information about the metagenome of each farm. - Do you mean "among" samples?
Response: The indicated error was corrected in line 135. - How many replicates were analyzed by t test?
Response: We are very sorry. The indicated sentence was deleted. - This text should be moved to section 2.1 (methods)
Response: The indicated sentence was moved to the section 2.1. of Materials and Methods. - Are the ± values SE? SD?
Response: The values are SE. - occupy? Do you mean represented? or comprised?
Response: The word was changed with “comprised” in line 156. - rate of viruses? Do you mean "prevalence based on the numbers of amplified reads"?
Response: The sentence was modified as “The results of the family-based taxonomic analysis showed that the relative abundance of viruses was approximately 41.2% of the metagenomic sequences identified at TO farm, followed by 15.0% at IS, 4.3% at BR, 4.0% at KB, and 1.6% at JH” in lines 168-170. - Virus families are written in italics.
Response: Changed as italics. - Table 1. Are these values prevalences (%)? Not stated.
Response: “Relative prevalence (%)’ was added in Tbale 1. - Genera are written in italics.
Response: Changed as italics. - Provide a reference here.
Response: The related references were added in line 199. - Are you specifically talking about insect guts here?
Response: The words were changed as “most insects” in line 209. - There are published studies on DNA viruses in scarab larvae and in other coleopteran larvae. Mention them. Reference 16 is not a useful source here.
Response: The indicated sentence was modified as “Eukaryotic DNA viruses that infect insects such as coleopteran species, orthopteran species, lepidopteran species and dipteran species mainly belong to the families Baculoviridae, Herpesviridae, Nudiviridae, Entomopoxviridae, Iridoviridae, Parvoviridae, and Asfarviridae” in line 224. The ref 16 suggests that insects may not harbor eukaryotic DNA viruses, and thus ref 16 was cited in this study because no specific eukaryotic DNA viruses were detected. - You performed this study and identified an abundance of phages, so what are your conclusions from the production standpoint? Are phages of benefit or detrimental to the production of scarab larvae?
Response: In this study, authors aimed to determine the composition of phage viruses and the presence of eukaryotic DNA viruses in PBL. Authors believe that more extensive studies are needed to describe the effect of phage composition on the productivity of scarab larvae. Our conclusion was described as “This study focused on profiling DNA viruses in the hindgut contents of PBL using metagenomic shotgun sequences. It was confirmed that more than 98% of DNA viruses in the hindgut were bacteriophages, mainly belonging to the Siphoviridae family. The detected eukaryotic DNA viruses were thought to likely have originated from contaminated feed or soil. These results provide a comprehensive understanding of the gut DNA viral composition of PBL and will aid in future studies of insect viromes” in lines 256-261. - Delete repetitive text. This is not a conclusion (its repeating preamble)
Response: The repetitive sentence was deleted. - This text is Discussion, not a conclusion.
Response: The indicated text was moved to the section of discussion after minor modification, and then the section of conclusion was deleted. - This text does look like a conclusion, but what are the implications for scarab production on a commercial scale?
Response: The balance of intestinal microorganisms is very important for the growth of insects. Viruses, especially bacteriophages, play an important role in the balance of intestinal microorganisms. Therefore, authors intend to use them to analyze differences in productivity among farms. Thus, the additional sentence was added as “These results provide a comprehensive understanding of the gut DNA viral composition of PBL and will aid in future studies of insect viromes” in lines 260-261. - Why is working for Hoxbio a potential conflict of interest? Does this company produce scarabs? Or what?
Response: The Hoxbio did not produce scarabs and looks good bacteria for increase of production. The author has only described in accordance with MDPI policy. - The references should be formatted following MDPI guidelines.
Response: The references were changed.
Other points
Supplemental Tables.
S1: are the farms in Korea? The BR diet is described as mushroom in the table, but is described as mushroom waste in the text. Which is it?
Response: The farms are located in Korea. Mushroom changed with “spent mushroom substrate” in Table S1.
S3. Indicate what the values mean? Are these percentages? Of what? The term "Microbial community " is not informative.
Response: The title was changed “The relative abundance (%) of microbiota of hindgut samples of P. brevitarsis larvae” in Table S3.
S4. Again - what do the values indicate exactly? "Taxonomy analysis" is not informative.
Response: The title was changed “The relative abundance (%) of hindgut viruses of P. brevitarsis larvae at the species level”
Round 2
Reviewer 3 Report
Comments and Suggestions for Authors
The authors have addressed most of my concerns.